

# Regional county-level housing inventory predictions and the effects on hurricane risk

Caroline J. Williams[1], Rachel A. Davidson[1], Linda K. Nozick[2], Joseph E. Trainor[3], Meghan Millea[4], Jamie L. Kruse[4].

[1]Department of Civil and Environmental Engineering, University of Delaware, Newark, DE, 19716, USA
[2]School of Civil and Environmental Engineering, Cornell University, Ithaca, New York, 14850, USA
[3]Biden School of Public Policy and Administration, University of Delaware, Newark, DE, 19716, USA
[4]Department of Economics, East Carolina University, Greenville, North Carolina, 27858, USA

Correspondence to: Rachel A. Davidson (rdavidso@udel.edu)

**Abstract.** Regional hurricane risk is often assessed assuming a static housing inventory, yet a region's housing inventory changes continually. Failing to include changes in the built environment in hurricane risk modeling can substantially underestimate expected losses. This study uses publicly available data and a long short-term memory (LSTM) neural network model to forecast the annual number of housing units for each of 1,000 individual counties in the southeastern United States over the next 20 years. When evaluated using testing data, the estimated number of housing units was almost always (97.3 %
of the time), no more than one percentage point different than the observed number, predictive errors that are acceptable for most practical purposes. Comparisons suggest the LSTM outperforms ARIMA and simpler linear trend models. The housing unit projections can help facilitate a quantification of changes in future expected losses and other impacts caused by hurricanes. For example, this study finds that if a hurricane with similar characteristics as Hurricane Harvey were to impact southeast Texas in 20 years, the residential property and flood losses would be nearly US$4 billion (38 %) greater due to the expected
increase of 1.3 million new housing units (41 %) in the region.

## 1 Introduction

Probabilistic regional hurricane risk assessments typically have been static, where the hazard is modeled as stationary and the built environment is considered to be unchanging. Recently, researchers have begun relaxing the former assumption as the effects of climate change on hurricane frequency and intensity are captured (Emanuel, 2011; Liu, 2014; Pant and Cha, 2018).
Nevertheless, changes in the building inventory over time have not received similar attention. The number, locations, and types of buildings exposed to hurricanes change continually over time in ways that can alter risk. In Harris County, Texas, home to Houston, for example, the population grew 36 % from 2000 to 2020 (U.S. Census Bureau, 2020a). Such a transformation could have a large effect on hurricane risk. If a risk assessment had been conducted in Harris County in 2000 based on the building inventory at the time, when there were 3.4 million residents living in 1.2 million housing units, it would have underestimated
the losses that occurred in Hurricane Harvey in 2017, by which time there were 4.5 million residents living in 1.7 million


housing units. Hurricane risk implications are especially notable for rapidly growing coastal counties such as Flagler County, Florida where the number of housing units has doubled since 2000, from 24,000 to 57,000 housing units.

Focusing on the number of housing units and their regional distribution by county (not changes in exact location or type), this
paper offers two contributions. First, using data for 1,000 counties in the Southeast United States from Texas to Delaware (Figure 1), a long short-term memory (LSTM) neural network model is developed to predict the number of housing units in each county over the next twenty years. LSTMs include feedback mechanisms for data in sequence and thus are well-suited for predictions on time series data. The LSTM model is evaluated through comparison to other model types commonly used for time series analyses, including a simple linear trend model and autoregressive integrated moving average (ARIMA) models.
Second, using the recommended new LSTM model, named the 20-Year Regional Annual County-Level Housing (REACH20) model, changes in the predicted number and distribution of housing units in the next twenty years are described and implications of those changes for hurricane risk are discussed.

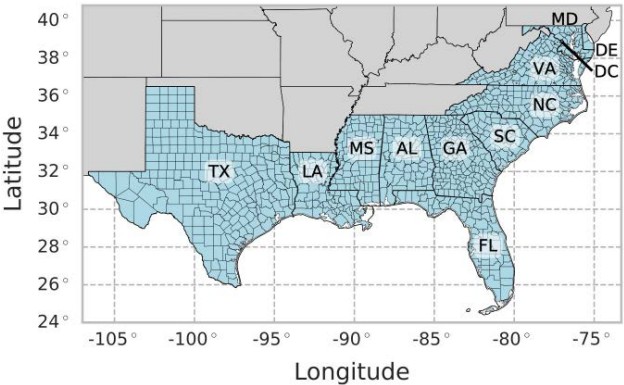

**Figure 1 Study area of 1,000 counties in the Southeast United States**

Following a review of related literature on land use change and housing change modeling in Sect. 2, the data and model types are described in Sect. 3 and Sect. 4, respectively. The set of specific analyses conducted are listed in Sect. 5 together with the metrics for evaluating and comparing the models. Results are presented in Sect. 6, including a comparison of the model types,
evaluation of the final recommended LSTM model, and discussion of the implications of projected change in the housing inventory. The paper concludes with a summary of the key findings and discussion of limitations and future work.

## 2 Literature review

Three bodies of literature support the proposed housing model, those focused on (1) regional land and population modeling, (2) housing economics, and (3) the intersection of natural hazards and the changing built environment.




## 2.1 Land use land cover change and population projections

The expansive land use land cover (LULC) change literature estimates physical changes to a landscape across a study region over time (Daniel et al., 2016; Sleeter et al., 2017). These models are used for a wide range of applications, such as evaluating urbanization trends or comparing ecosystem conservation approaches, and often model changes in land dynamics over a long

period of time usually at decadal intervals. The units of analysis are typically at a 1 km$^2$ or less and can span a regional (multi-county) area. There are three predominant methods for LULC modeling for a large spatial scale: machine learning (ML), cellular automata (CA), and a combination of ML and CA. While ML methods use historical land use data to predict land use change behavior, CA methods develop localized land use or land cover transition maps with neighborhood transition rules over a uniform grid to predict how the land use or land cover in a grid cell will change over time (National Research Council,

2014). Aburas et al. (2019), Briassoulis (2019), Musa et al. (2017), and Verburg et al. (2004) provide reviews of different ML and CA methods for LULC modeling as well as commonly used model parameters. In the common combination methods, ML is often used to calibrate the weighting for land use transition maps and CA is used to define local rules for land use transition (Aburas et al., 2019). In recent years, deep learning neural network methods for LULC modeling have developed substantially, where convolutional neural networks (CNN) perform well for a study of spatial dynamics at a point in time, recurrent neural

networks (RNN) work well for time-series data for a single location, and a combination of the two methods, ConvLSTM, incorporates both spatial and temporal data (Cao et al., 2019; Ienco et al., 2017; Ye et al., 2019).

Population projection models estimate the number of people residing in an area over a series of time steps in the future. While most population projections are developed with a unit of analysis at a country or state level (University of Virginia, 2018; U.S.

Census Bureau, 2017), one population projection dataset developed by Hauer (2019) uses the Hamilton-Perry method (Swanson et al., 2010) to estimate population changes for all U.S. counties at five-year intervals between 2020 and 2100 for eighteen age groups, two sex groups, and four race groups under five climate change scenarios. Assuming the amount of urban land cover and infrastructure is proportional to the number of people within an area, population estimates are commonly used as a metric for a society's exposure to risk (Tellman et al., 2021; Wing et al., 2018).


While the LULC models and population projection models aim to represent physical and demographic changes over many years across a region, little work has studied the changes in regional housing dynamics specifically. This study aims to address this gap in the literature.





## 2.2 Housing economics

The urban economics, real estate, and housing literature examine the theorized drivers of housing development. Researchers largely agree that drivers of real estate cycles are rooted in economic fundamentals, such as local supply and demand and Urban Growth Theory (Edelstein and Tsang, 2007; Mayer and Somerville, 2000). Computable general equilibrium (CGE) as well as supply and demand land value models are especially common in the housing market literature and can be applied from a local to country spatial scale (Ali et al., 2020; Cho et al., 2005; Ustaoglu and Lavalle, 2017). Modeling methods also include system dynamics and agent-based modeling (ABM) approaches, which capture the interaction between individual decision-making and economic effects at a local scale (Filatova, 2015; Magliocca et al., 2011; Wheaton, 1999). The spatial and temporal scales of economic and housing models ultimately depend on the degree of detail for change interaction (such as agent decisions), the amount of data available, and the study point of interest. However, none of the models reviewed incorporated the explicit spatial component of annual changes in housing units across a region at a county-level over time.

## 2.3 Exposure to natural hazards over time

There is a limited group of studies that evaluate a society's changing exposure to natural hazard risk over time. Davidson and Rivera (2003) use population projections and headship rate data to predict the number, location, and types of housing units per census tract in a region at 5-year intervals between 2000 to 2020. The results were later used in a hurricane risk study for North Carolina (Jain and Davidson, 2007). Chang et al. (2019) studied the effect of urban development patterns on future flood risk or earthquake risk in the Vancouver regions for the year 2041 under three prescribed development scenarios—status quo, compact, and sprawl. Song et al. (2018) compared three ML methods to predict the land use change in Bay County, Florida in 2030 and evaluated the risk due to sea level rise under two growth rates and two policy scenarios. Hauer et al. (2016) also used a modified version of the Hammer method (Hammer et al., 2004) to predict the number of people at risk of sea level rise per census block, based on decadal housing estimates for the coastal areas of the continental United States, between 2010 and 2100 under five development scenarios. Sleeter et al. (2017) used a CA model to evaluate changes in land cover and the effect on tsunami risk in the U.S. Pacific Northwest at annual increments between 2011 and 2061. Keenan and Hauer (2020) compared 30-year population projections in Puerto Rico with planned hurricane recovery and resiliency investments, finding an overestimation of future fiscal and infrastructure needs compared to the projected decline in population. Strader et al. (2015) used the CA-based Spatially Explicit Regional Growth Model (SERGoM) (Theobald, 2005) to model changes in housing density, alongside the U.S. EPA's Integrated Climate and Land Use Scenarios (ICLUS) to review the volcanic hazard exposure in the Northwest U.S. at a decadal scale between 2010 and 2100 under five scenarios. Similarly, Freeman and Ashley (2017) use SERGoM to review hurricane risk in the U.S. for the same time interval under two hurricane scenarios. While these studies have all considered the changes in a society's population, building inventory, or land uses and the effects on natural hazard



risk, none include the annual changes in housing units at a county-level over a forecast period across a multi-state region in the U.S.

## 2.4 Predictor variables

An important piece of developing the proposed housing model in this paper is understanding the theorized predictors of land use change, population change, and housing development among the different bodies of work reviewed. Table 1 summarizes the 32 predictors that emerged from the literature as important predictors of housing inventory changes. The data sources for each component in Table 1 are available in Table S1 (Sect. S1.3 of Supplemental Section). Section 3 describes the data selection methodology used for the proposed model.


**Table 1 Predictors of housing inventory changes over time**

| Category | Predictors[1] | Category | Predictors[1] |
|---|---|---|---|
| Population | Population[2] | Demographics | Race[2] |
|  | Population density[2] |  | Age[2,3] |
|  | Migration[2,3] |  | Marital status[2,3] |
| Housing | Housing units[2,3] |  | Education[2,3] |
|  | Housing density[2,3] | Land | Land cover[2] |
|  | Single family housing units[2,3] |  | Land use |
|  | Single family housing unit density[2,3] |  | Available buildable land[2] |
|  | Vacancy rate[2,3] |  | Proximity to coastline[2] |
|  | Owner–occupancy rate[2,3] | Economic | Property value[2,3] |
|  | Household size[2,3] |  | Land value |
|  | Lot size |  | Property tax rate[2,3] |
|  | One–unit building permits[2] |  | Mortgage interest rate |
|  | Year built[2,3] |  | Construction cost |
|  | Householder tenure[2,3] |  | Cost of living |
|  | Household income[2,3] |  | GDP[2] |
|  |  |  | GINI index |
|  |  |  | Employment rates[2] |
| [1]Data sources for all predictors are available in Table S1 [2]Denotes it was considered in the proposed model. [3]Denotes data only available on a decadal basis prior to 2010 (Fig. S1). | | | |

## 3 Data

Modeling the annual changes in the number of housing units for 1,000 counties over a 10-, 20-, or 30-year time horizon requires a dataset of annual county-level data for more than 10 years for all counties in the study area. Of the 32 predictors identified

as potential predictors of new housing construction, only 25 (indicated by "[2]" in Table 1) had county-level data available for

more than 10 years and were considered for this study. Data for these 25 predictors were compiled into a dataset for all available years from 1970 on (Sect. S1 of Supplemental Section). Data for 16 predictors are only available on a decadal basis prior to 2010 (indicated by "³" in Table 1), requiring linear interpolations to provide a consistent annual dataset. Figure S1 (Sect. S1.4 of Supplemental Section) details which years data are available for each predictor. Of the 25 predictors considered, 19 have

data available starting in 1990 or earlier. Lastly, due to the significant impact of the Great Recession on the nation's housing construction industry, data in 2008, 2009, and 2010 were removed. While the impact of shocks, such as the Great Recession or the COVID-19 Pandemic, have caused sizeable disruptions to the housing market and should be considered in resiliency planning, the goal of this work is to predict the number of net new housing units under normal conditions. Predicting economic shocks is outside the scope of this work. In total, the individual variables used for this study are available in time intervals

from 16 years (2001–2007, 2011–2019) to 46 years (1971–2007, 2011–2019) for 1,000 counties. For details about the data preprocessing, see Sect. S1 of the Supplemental Section.

## 4 Model types

To estimate the number of new housing units per county over the next 10 to 30 years across a region, a set of time series models and range of model parameters were considered. The time series models tested include a simple linear trend model,

autoregressive integrated moving average (ARIMA) models, and long-short term memory (LSTM) neural network models. The ranking criteria for all models compared in this study was prediction performance of the number of housing units for 30 years in the future. Linear trend and ARIMA models were tested because of their ease of use, common application across disciplines, and interpretability, while neural network models were considered for their ability to handle large quantities of spatial and temporal data.


### 4.1 Linear trend

The simple linear trend method consisted of fitting one univariate linear model to each county using ordinary least squares (OLS) regression. Each model was fit to the number of housing units and the resulting trend line was extrapolated to estimate the number of housing units for the following 10, 20, and 30 years.


### 4.2 Autoregressive Integrated Moving Average (ARIMA)

ARIMA models are univariate linear models that use lagged observations of the time series data and are the most common methods for time series modeling (Box et al., 2016). Equation (1) presents an ARIMA model to predict the value of variable $y$ at time $t$ as a function of values of $y$ at previous time steps ($y_{t-1, ..., } y_{t-p}$) and error terms at time $t$ and at previous time steps

($\epsilon_t, \ldots, \epsilon_{t-q}$). The parameters $\alpha$, $\beta_1$, …, $\beta_p$, and $\phi_1$, …, $\phi_q$ are estimated from the data. ARIMA models are typically referred



to by the values (p, d, q), where p is the number of lags for the autoregressive term, d is the number times the data must be differenced to be stationary prior to model fitting, and q is the number of lagged forecast errors for the moving average term. This study also compares the method of using one ARIMA model for all counties in the study area (one set of p, d, and q values), versus an individual ARIMA model for each county, to understand whether a simple uniform ARIMA model could

be used across the study region. The annual percent change in number of housing units was used as y in Eq. (1).

$$y_t = \alpha + \beta_1 y_{t-1} + \beta_2 y_{t-2} + \cdots + \beta_p y_{t-p} + \phi_1 \epsilon_{t-1} + \phi_2 \epsilon_{t-2} + \cdots + \phi_q \epsilon_{t-q} + \epsilon_t \tag{1}$$

### 4.3 Long Short-Term Memory (LSTM)

Neural network models have emerged as a common method for analyzing complex problems due to their ability to handle

large, nonlinear datasets with high accuracies. Recurrent neural networks (RNN) are specifically utilized for sequential modeling applications, such as time series forecasting and natural language processing, and can be used to predict future housing inventories given a sequence of variables with nonlinear relationships across a large study area. LSTM models are the most common among the family of RNNs available and were chosen in this study for their ability to learn both long-term and short-term dependencies across a sequence of multivariate input data. The time dependencies are learned in an LSTM unit

across a series of LSTM memory cells. Each cell consists of three "gates" that manage the information passed across the sequence of input data. The "input gate" regulates whether to add new information to the memory of the cell, the "forget gate" removes information to be considered in the given memory cell, and the "output gate" regulates the information leaving the cell. For more on LSTM models, see Hochreiter and Schmidhuber (1997), Ienco et al. (2017), and Wang et al. (2020b).

All neural network models, including LSTMs, have a set of hyperparameters that are unique to a given model and are tuned to improve model performance. For LSTM models, tuning parameters include the number of input time steps and output time steps, number of feature and targets, number of layers and nodes, activation method, loss metrics, type of optimizer, learning rate, batch size, batch normalization, use of dropouts and dropout rates, and number of epochs. Data is also split into training and testing sets typically using a 70/30 or 80/20 ratio, allowing a model's performance to be evaluated both on the data for

which it is developed (the training set) and an independent data set (the testing set). Lastly, due to variability in each run of the neural network algorithm, a single model configuration is often tested multiple times to search for the model producing the lowest errors.

### 5 Analyses and evaluation

To identify the best time series model for predicting the number of housing units up to 30 years in the future, a range of model

configurations was tested (Table 2). The set of feature variables (also known as independent or explanatory variables) and the target variable (also known as dependent or response variable) for each model is provided in Table 3. The target variable for




the linear trend model is $h_{itk}$, the *number* of housing units for county $i \in (1,\dots,I)$ in year $t \in (1,\dots,T_o)$ in sample $k \in (1,\dots,K)$, where a sample $k$ is one sequence of input and output years for county $i$ (Fig. 2). The target variable for remaining models is $r_{itk}$, the annual percent *change* of the number of housing units for county $i$ in year $t$ in sample $k$, defined in Eq. (2).


$$r_{itk} = \frac{h_{itk} - h_{i,t-1,k}}{h_{i,t-1,k}} * 100 \tag{2}$$

**Table 2 Model tests**

| Test | Model type[2] | Feature and target Set[1] | Input length (years), $T_i$ | Output length (years), $T_o$ | Number of samples, $K$ | Spatial weighting |
|---|---|---|---|---|---|---|
| A1 | Linear trend | I | | | | |
| A2 | ARIMA | II | 6 to 36 | 10 | 31,000 to 1,000 | No |
| A3 | LSTM | II | | | | |
| B1 | Linear trend | I | | | | |
| B2 | ARIMA | II | 6 to 26 | 20 | 21,000 to 1,000 | No |
| B3 | LSTM | II | | | | |
| C1 | Linear trend | I | | | | |
| C2 | ARIMA | II | 6 to 16 | 30 | 11,000 to 1,000 | No |
| C3 | LSTM | II | | | | |
| D | LSTM | III | 6 to 36 | 10 | 31,000 to 1,000 | No |
| E | LSTM | III | 6 to 26 | 20 | 21,000 to 1,000 | No |
| F | LSTM | III | 6 to 16 | 30 | 11,000 to 1,000 | No |
| G | LSTM | IV | 6 to 17 | 10 | 12,000 to 1,000 | No |
| H | LSTM | IV | 6 to 7 | 20 | 2,000 to 1,000 | No |
| I | LSTM | III | 6 to 26 | 20 | 21,000 to 1,000 | Yes |

[1]Feature and target sets are defined in Table 3
[2]For Tests A3, B3, and C3, for each

**Table 3 Feature and target sets**

| Feature and target set | Years available | Total years | Features | Target |
|---|---|---|---|---|
| I | 1971–2007, 2011–2019 | 46 | • Number of housing units ($h_{itk}$) | • Number of housing units ($h_{itk}$) |
| II | 1971–2007, 2011–2019 | 46 | • Annual percent change in number of housing units ($r_{itk}$) | • Annual percent change in number of housing units ($r_{itk}$) |
| III | 1971–2007, 2011–2019 | 46 | • Population<br>• Population density (person per km²)<br>• Number of housing units<br>• Housing unit density (units per km²)<br>• Percentage of vacant housing units<br>• Percentage of owner-occupied housing units<br>• Average household size<br>• Percentage non-white population<br>• Percentage of population with high school degree<br>• Percentage of population with college degree<br>• Percentage of non-buildable land area | • Annual percent change in number of housing units ($r_{itk}$) |



| | | | | |
|---|---|---|---|---|
| | | | • Distance to coastline (m)<br>• Annual percent change in number of housing units ($r_{itk}$) | |
| IV | 1990–2007, 2011–2019 | 27 | Everything in Feature Set II and:<br>• Number of one-unit detached housing Units<br>• One-unit detached housing unit density (units per km²)<br>• Percentage of one-unit detached housing units of total housing units<br>• One-unit detached housing units per capita<br>• Number of one-unit housing building permits<br>• Number of one-unit housing building permits per number of one-unit detached housing units<br>• Median household income (USD)<br>• Median age<br>• Percent of married population<br>• Median property value (USD)<br>• Number of jobs<br>• Jobs per capita | • Annual percent change in number of housing units ($r_{itk}$) |

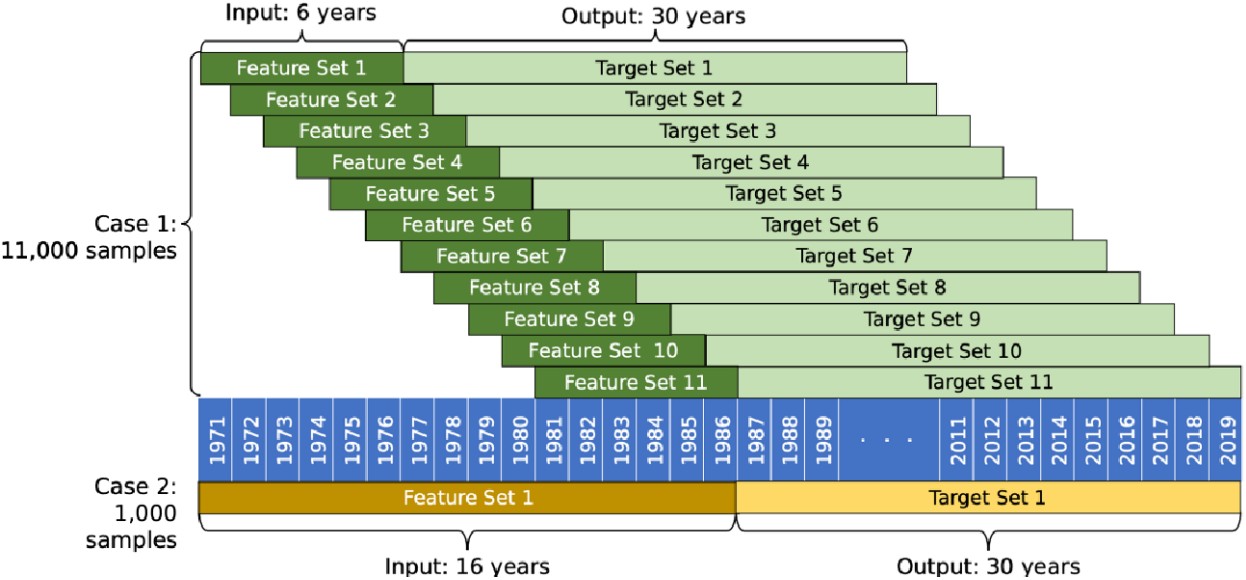

**Figure 2 The change in sample size ($K$) for two different input year lengths ($T_i$) for Test C1**

The two target variables, $h_{itk}$ and $r_{itk}$, are directly related, but the range of $h_{itk}$ values for all counties across all available years spans multiple orders of magnitude, from 50 to 1.8 million housing units. The large spread in the data makes it difficult to fit a model across all counties and all years with $h_{itk}$ as a target variable. The use of $r_{itk}$ overcomes this problem, with values from –78 % to 132 % annual change in housing units.

Each test predicts values for all 1,000 counties over a 10-, 20-, or 30-year time period so that a model with a 30-year projection period, for example, predicts 30,000 unique county-year values. A range of input sequence lengths were compared across all tests to determine the optimal input and output length structure for each model type. The combined input and output lengths determine the total number of samples, $K$, used to train and test the model, where a shorter time interval leads to more samples for training and testing a model, while a longer time interval leads to fewer samples for the model. Specifically, when the sum

of the input and output length ($T_i+T_o$) is less than the number of years in the data set, $T$, there are ($T–(T_i+T_o)+1$) samples of data for each county. As an example, say Test C3 was implemented for just one county. Test C3 uses Feature Set I, which has $T$=46 years of available data and a $T_o$=30-year output length. For one case evaluated in Test C3 that has an input length of $T_i$=6 years, the resulting total input/output time interval is 36 years, leading to a total of 46–(6+30)+1=11 different time intervals across the 46 years of available data for the single county. For all 1,000 counties in the study area, this test configuration would

result in $K$=11,000 samples available to train and test the model (Case I, Fig. 2). However, if the input length is instead $T_i$=16 years and the output length is $T_o$=30 years, the total length of the time interval is 46 years, allowing only 46–(16+30)+1=1 sample for a given county and $K$=1,000 samples over the entire study area (Case II, Fig. 2).

In Tests A, B, and C, the univariate linear trend, ARIMA, and LSTM models were compared to identify the best input/output

length combination for each model and the best univariate model performance. Since the linear trend and ARIMA models are restricted to one variable, for fair comparison, the LSTM was similarly restricted in Tests A, B, and C. These tests used data available since 1971, thus providing 46 years of data to fit the model (note that the Great Recession is excluded). For the simple linear trend modeling, each county was fit to an individual linear model and errors were aggregated across all counties. Similarly, the ARIMA models fit individual ARIMA models for each county for a given $p$, $d$, and $q$ combination and errors

were aggregated across all counties. The $p$, $d$, and $q$ values tested ranged from 0 to 2. For LSTM models in Tests A, B, and C, the best of 5 LSTM runs for each input/output combination was taken as the solution.

Tests D, E, and F compared the multivariate LSTM models to identify the best input/output length combination for each model and the best multivariate model performance. These tests only included the 13 feature variable in Feature Set III which were

available since 1971 and provided 46 years of available data. LSTM models in Tests D, E, and F recorded the best of 10 LSTM runs.

Tests G and H used LSTM models with 25 feature variables to understand whether more features improve model performance. A tradeoff exists between including more features, but having a shorter timespan of available data, and including fewer

features, but having a longer timespan of available data. Feature Set IV used in Tests G and H is only available since 1990 and provides just 27 years of data. These two tests recorded the best of 10 LSTM runs.



The literature suggests there are both time and space dependencies when modeling housing projections (Cho et al., 2005; Strader et al., 2015), thus Test I reviewed an LSTM model that included spatial weighting across all counties for all features

in Feature Set III. With influence from graph neural network methods (Wu et al., 2021), spatial weighting was applied so that feature values in each county were averaged among all contiguous counties prior to model fitting. For example, the population feature variable for a given county would be reassigned as the non-weighted average population value of the county itself and all counties directly adjacent. The values for the remaining feature variables for a given county would then be similarly reassigned. Once spatial weighting was applied to all counties for all feature variables, then the model was fit accordingly. No

spatial weighting was applied to the target variable and this test recorded the best of 10 LSTM runs.

For all LSTM models in Tests A through I, samples were randomly divided for a given input/output combination into a training and testing set using an 80/20 split. As a result, the set of samples for a given county were randomly distributed into the training and testing sets. Holdout validation was not implemented in this study because the developed model is not intended for use

outside the defined study area of 1,000 counties. Both training and testing errors are tracked to identify possible overfitting. The same hyperparameters were used in all LSTM models and are provided in Table S2 (Sect. S2.3.2 of Supplemental Section).

All models were evaluated using the root mean squared error, $RMSE_r$, of the annual percent change of housing units (Eq. (3)), as well as the expected value, $E[|H|]$, and standard deviation, $s_{|H|}$, over all $I$, $T_o$, and $K$, of the absolute value of the percent

relative error in number of housing units, $H_{itk}$ (Eq. (4)), where $\hat{r}_{itk}$ is the predicted annual percent change of housing units in county $i$, year $t$, and sample $k$; $r_{itk}$ is the observed annual percent change of housing units; $\hat{h}_{itk}$ is the predicted number of housing units, $h_{ikt}$ is the number of observed housing units; and $I$, $T_o$, and $K$ are the numbers of counties, number of years in the output series, and number of samples, respectively.

$$RMSE_r = \sqrt{\frac{\sum_{ikt}(\hat{r}_{itk}-r_{itk})^2}{IT_oK}} \tag{3}$$

$$H_{itk} = \frac{\hat{h}_{itk}-h_{itk}}{h_{itk}} * 100 \tag{4}$$

The $RMSE_r$ is based on the target value optimized by the LSTM and the response variable for the ARIMA, $r_{itk}$; the $E[|H|]$ and $s_{|H|}$ are included because they are based on the more easily interpreted variable $h_{itk}$. The linear trend and ARIMA models

do not separate the data into a training and testing set, therefore the errors were calculated across all output years in all samples. Of the multiple input/output lengths evaluated for each test, and the multiple runs for the LSTM models, the input/output combination with the lowest multiplied of $RMSE_r$, $E[|H|]$, and $s_{|H|}$ values for each test is reported. For example, in Test A3, where 6 to 36 input years were evaluated and the output length was 10 years, there were 36–6+1=31 different models evaluated,





each over five runs. Of those 31*5=155 models, the model with the lowest average $RMSE_r$, $E[|H|]$, and $s_{|H|}$ value was

reported as the best model for Test A3.

Each time series model was fitted and evaluated using a publicly available Python (Van Rossum and Drake, 2009) library—Scikit-Learn package for the linear trend model (Buitinck et al., 2011), Statsmodel package for ARIMA (Seabold and Perktold, 2010), and Tensorflow package for LSTM models (Martín Abadi et al., 2015).

**6 Results**

**6.1 Model comparison**

**6.1.1 Model type comparison**

We first compare the model types. For the univariate models evaluated in Tests A, B, and C, the LSTM method outperforms the simple linear trend and ARIMA models for 10-, 20-, and 30-year prediction periods (Table 4). For the 30-year prediction

period, for example, the linear trend, ARIMA, and LSTM models have $RMSE_r$ values of 2.0, 1.8, and 1.2, respectively and $E[|H|]$ values of 11.4, 12.7, and 0.64, respectively (Table 4, Tests C1, C2, C3).

**Table 4 Results**

| Test[1] | Model type | Set[2] | Parameters | $T_i$ | $T_i+T_o$ | K | Spatial weighting | $RMSE_r$ (%) | $E[|H|]$ (%) | $s_{|H|}$ (%) |
|---|---|---|---|---|---|---|---|---|---|---|
| A1 | Linear trend | II | N/A | 9 | 10 | 1,000 | No | 1.883 | 4.199 | 5.329 |
| A2 | ARIMA | I | p,d,q = 1,0,1 | 16 | 10 | 21,000 | No | 1.433 | 3.637 | 5.118 |
| A3 | LSTM[3] | I | LSTM hp[4] | 21 | 10 | 16,000 | No | 0.961 / 1.161 | 0.484 / 0.558 | 1.608 / 1.31 |
| B1 | Linear trend | II | N/A | 6 | 20 | 1,000 | No | 1.759 | 6.220 | 7.727 |
| B2 | ARIMA | I | p,d,q = 1,0,1 | 14 | 20 | 13,000 | No | 1.643 | 8.390 | 12.049 |
| B3 | LSTM | I | LSTM hp | 11 | 20 | 16,000 | No | 1.497 / 1.084 | 0.626 / 0.623 | 2.64 / 1.046 |
| C1 | Linear trend | II | N/A | 16 | 30 | 1,000 | No | 1.997 | 11.432 | 12.234 |
| C2 | ARIMA | I | p,d,q = 1,0,1 | 6 | 30 | 11,000 | No | 1.767 | 12.704 | 18.648 |
| C3 | LSTM | I | LSTM hp | 11 | 30 | 6,000 | No | 1.388 / 1.164 | 0.556 / 0.644 | 2.482 / 1.014 |
| D | LSTM | III | LSTM hp | 21 | 10 | 16,000 | No | 0.484 / 0.636 | 0.209 / 0.287 | 0.947 / 0.869 |
| E[5] | LSTM | III | LSTM hp | 11 | 20 | 16,000 | No | 0.195 / 0.426 | 0.116 / 0.196 | 0.406 / 0.557 |
| F | LSTM | III | LSTM hp | 11 | 30 | 6,000 | No | 0.534 / 0.781 | 0.107 / 0.254 | 0.917 / 0.804 |
| G | LSTM | IV | LSTM hp | 12 | 10 | 6,000 | No | 1.531 / 1.134 | 0.326 / 0.501 | 4.015 / 1.243 |
| H | LSTM | IV | LSTM hp | 7 | 20 | 1,000 | No | 0.098 / 1.384 | 0.122 / 0.708 | 0.515 / 1.413 |
| I | LSTM | III | LSTM hp | 11 | 20 | 16,000 | Yes | 0.303 / 0.495 | 0.138 / 0.279 | 0.519 / 0.643 |

[1]Of the multiple input/output lengths evaluated for each Test, and the multiple runs for the LSTM models, the input/output combination with the lowest average of $RMSE_r$, $E[|H|]$, and $s_{|H|}$ values for each test is reported.
[2]Feature and target sets are defined in Table 3
[3]All LSTM models report Training errors / Testing errors
[4]See Table S2 for a list of the hyperparameters used for all LSTM models
[5]Recommended REACH20 model





Comparing the linear trend and ARIMA models, the best model type depends on the metric used and output length. In terms of $RMSE_r$, the ARIMA performs better than linear trend models for all output lengths. In terms of $E[|H|]$, however, the linear trend model is 2.17 and 1.27 percentage points better than the ARIMA for 20- and 30-year output lengths, respectively. The error distribution for the linear trend and ARIMA models are nearly the same (Fig. 3a, Fig. 3b, and Fig. 3c). Therefore, when quick long-term projections are needed, a simple linear trend model method may be adequate. The distribution of the testing errors for the LSTM model is much smaller than for linear trend and ARIMA models, and all output lengths have a similar distribution (Fig. 3d).

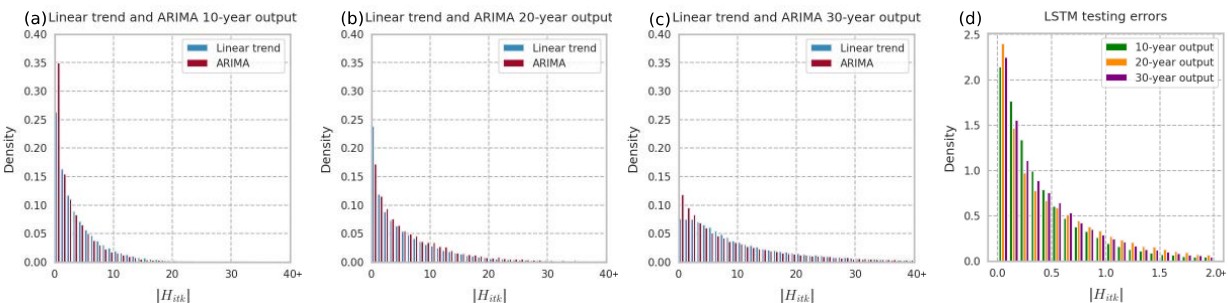

**Figure 3 Absolute percent relative error ($|H|_{itk}$) distributions for univariate models. (a) Linear trend vs. ARIMA 10-year output; (b) Linear trend vs. ARIMA 20–year output; C) Linear trend vs. ARIMA 30-year output; (d) LSTM for all output lengths (Note x-axis scale for (d) differs from the others.)**

### 6.1.2 Input and output lengths

A key issue in fitting these models is determining the best number of years of input and output data to use. The number of years of output, $T_o$, will depend in general on the intended use of the model, although it may be important to understand the tradeoff between forecasting for a longer duration into the future and keeping errors lower in case there is flexibility on the required output length. The results suggest that, as expected, errors in terms of $E[|H|]$ are larger for longer output lengths (Tests A, B, C, Table 4). That is, it is easier to forecast the number of housing units accurately for 10 years than for 20 years, and easier to forecast 20 years than 30. For errors in terms of $RMSE_r$, the pattern is similar, though not as consistent.

For a specified desired output length, the optimal number of years of input is not obvious a priori, as it depends on data availability and the extent to which variable values from previous years help predict target variable values in future years. If the value of a variable $x$ in each year $t$ is related to that in the preceding year $t–1$, then $x_{t–2}$ has an implicit indirect effect on $x_t$ as well, through $x_{t–1}$. Thus, it may be that including data for $x$ from many input years helps predict $x_t$, but it may not be required, and could just add noise. The housing vacancy rate in 1970 may not be relevant to the change in the number of housing units from 2020 to 2021, for example, beyond the indirect influence it has on the changes in the intervening years. The input length



also affects the total number of samples available to fit a model, where there is a tradeoff between a longer input length and fewer total samples versus a shorter input length with more training samples (Fig. 2).

The best-performing linear trend models all had input sequence lengths shorter than the output sequence lengths. With 46 years of data total, when the output length is 10 years, for example, the maximum input length is 36 years, but the best linear trend model had an input length of 9 years Table 4, Test A1).

For the ARIMA models, shorter input lengths performed better, where 16, 14, and 6 years were identified as the best input
lengths corresponding to 10, 20, and 30 years of output (corresponding to 21,000, 13,000, and 11,000 available samples, respectively). Additionally, for all output lengths, the best $p$, $d$, $q$ values tested were 1, 0, 1 respectively, suggesting that just one lag of the autoregressive term, one lag of error terms, and no differencing for the annual percent change of housing units data can be used for quick and approximate housing forecasts.

The univariate and multivariate LSTM models have the same best input length for a given output length, where the best input lengths include the years in either one decade (11 years inclusive) or 2 decades (21 years inclusive). This could result from the nature of the data availability, where most variables are only available at a decadal scale prior to 2010 (Fig. S1, Sect 1.3 of Supplemental Section).

**6.1.3 LSTM model comparisons**

Focusing on the LSTM models, which offer the smallest errors, we investigate feature selection, spatial weighting, and possible overfitting. To determine if additional feature variables help forecast the number of housing units in each county, we compare models that are the same except for the feature set. Test A3, B3, and C3 use feature set I (only the target variable); Test D, E, and F use feature set III (13 additional feature variables); and Test G and H use feature set IV (with another 12 additional
feature variables) (Table 3). The multivariate LSTM models in Tests D, E, and F outperform the univariate LSTM models evaluated in Tests A, B, and, C on all metrics and for all output lengths, where the errors from the multivariate model are approximately half those from the univariate model (Table 4). This suggests that the feature variables in set III do substantially improve prediction of future numbers of housing units. Comparing Tests D and E to Tests G and H, however, indicates that incorporating the additional 12 feature variables of feature set IV does not improve prediction. Since data are only available
since 1990 for variables in feature set IV, there is a tradeoff between adding the features and maximizing the duration of data availability, and the results suggest incorporating the additional features does not add value to the modeling.

Of all the LSTM models evaluated in Tests A through H, the best performing model is Test E, a multivariate LSTM having 11 input years and 20 output years with 13 features of data that are available since 1971. When, in Test I, spatial weighting was

added to the features for the same 11-year input, 20-year output model, there was no substantial improvement in errors. The test data $E[|H|]$ for Test E (without spatial weighting) and I (with spatial weighting) are 0.196 and 0.279, respectively.

Finally, comparing the testing and training errors for all LSTM models and both $RMSE_r$ and $E[|H|]$, does not suggest a substantial overfitting or underfitting problem. Across the 9 LSTM models, the median value of the ratio

$RMSE_r$(testing)/$RMSE_r$(training) is 1.71, and the median of $E[|H|]$(testing)/$E[|H|]$(training) is 1.54.

Based on all the results in Table 4, the best LSTM model in Test E is considered the recommended model to predict the number of housing units, $h_{itk}$, for the 1,000 counties in the study area over a 20-year period. This model is henceforth referred as the 20-Year Regional Annual County-Level Housing (REACH20) model. If an application required a 30-year prediction period,

the best LSTM model in Test F, with 11 input years and 30 output years, would be recommended.

## 6.2 Evaluation of recommended LSTM model

This section evaluates the recommended the 11-year input and 20-year output multivariate LSTM REACH20 model in more detail, examining the magnitude and distribution of errors. The REACH20 model has an expected absolute percent relative

error ($E[|H|]$) for the testing set of less than 0.2 % when comparing the predicted number of housing units, $h_{itk}$, with the observed number of housing units, $\hat{h}_{itk}$. That means, on average, across all predicted years $t \in T_o$, samples $k$, and counties $i$, the number of housing units predicted differs from the actual number of housing units by less than 0.2 %, likely negligible for many applications. Additionally, of the 64,000 predicted data points in the testing set (3,200 samples in the testing set (16,000*0.2) and 20 predicted years), almost all (97.3 %) had absolute percent relative errors of less than one percent

($|H_{itk}|<1.0$). Figure 4a provides a distribution of the relative errors among the predicted data points, with no bias and an even balance of over- and under-prediction.

When reviewing the variability of the testing set errors over the duration of the 20-year prediction period, the expected value $E_t[|H|]$ over all counties $i$ and samples $k$ of the absolute value of the percent relative error for each time step $t$ for the testing

set remains under 0.5 %. There is a noticeable, gradual increase in the errors as the predicted year horizon expands. The $E[|H|]$, for example, is 0.12 % in the first time step and 0.47 % in the twentieth time step (Fig. 4b). This suggests that while the errors are quite low for all years in the 20-year prediction period, the model does not predict the number of housing units 20 years in the future as well as it does the number of housing units one to five years in the future, as expected.



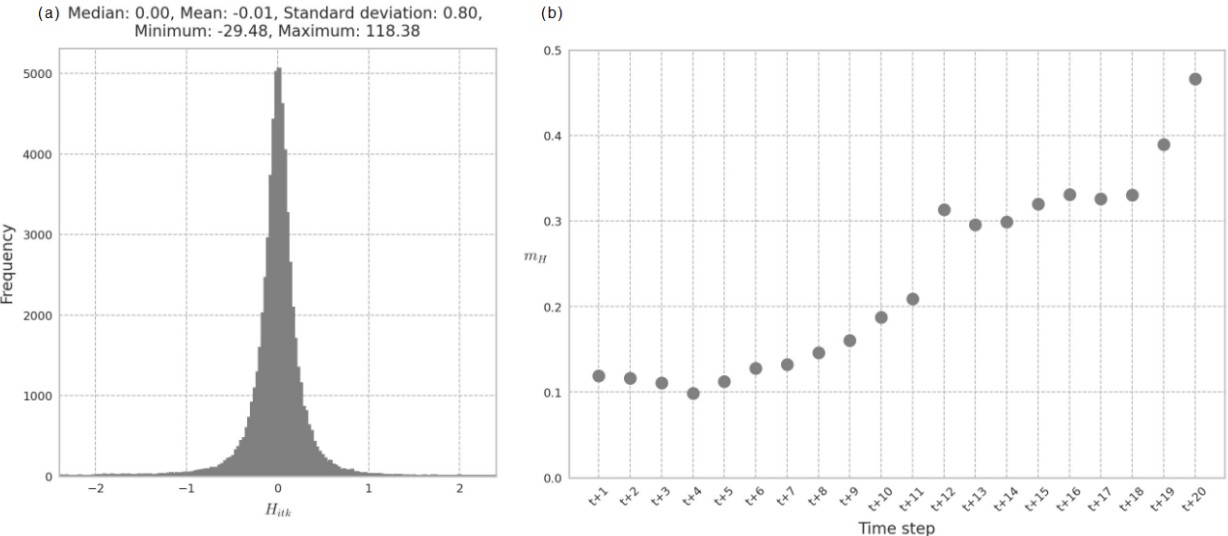

**Figure 4 (a) Distribution of $H_{itk}$ of the testing set using the REACH20 model (b) Average absolute percent relative error for each timestep over all counties and samples, $E_t[|H|]$, for the testing set using the REACH20 model (Eq. (7))**

Errors across space were also reviewed to understand whether the model performs better for certain geographic areas (e.g., urban vs. rural counties, or East coast vs. Gulf coast). Figure 5 shows the spatial distribution of the expected value $E_i[|H|]$ over all predicted time steps $t \in To$ and samples $k$ of the absolute value of the average percent relative error for each county $i$ for the combined training and testing set. Given that the samples for a given county and time sequence were randomly split into the training and testing set, a spatially complete view of the errors required a combination of the training and testing errors. There is no obvious spatial pattern across the study area, with a balance of over-prediction (purple) and under-prediction (green) across the region. There were 987 counties (98.7 %) with averaged $H_{itk}$ across all time steps less than 0.2 %. Errors in West Texas are slightly larger than other regions of the study area perhaps due to the relatively small population of these counties and the associated sensitivities to small changes in the error.

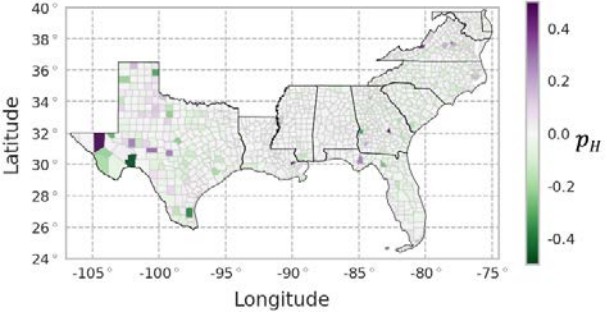

**Figure 5 Average percent relative error for each county over all predicted time steps and samples, $E_i[|H|]$, for the combined training and testing set using the REACH20 model (Eq. (8))**





### 6.3 Implications of projected change in housing inventory

Over the entire study region, the REACH20 model predicts approximately 16.7 million more homes in the 20-year forecast period between 2019 and 2039 (38 % growth, Fig. 6). The county-level projections aggregated in Fig. 6 are based on the last
11 years of available data excluding the Great Recession (2006, 2007, 2011–2019) for the 13 included features to estimate 20 years of future housing unit projections across all 1,000 counties using the REACH20 model.

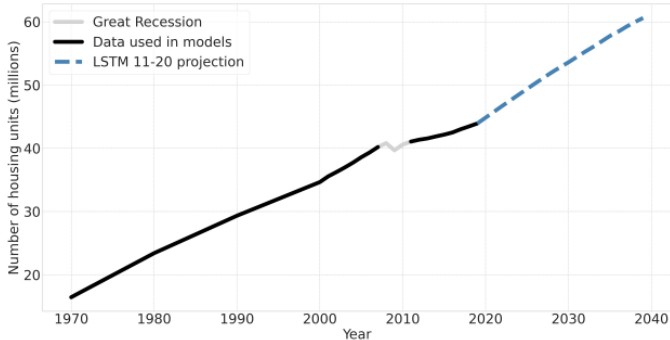

**Figure 6 Predicted number of housing units in the study area using the REACH20 model**

The projected housing rates of change across all counties over 20 years (2019–2039) vary spatially, where the housing inventory in almost all counties (97.4 %) is expected to grow over the next 20 years, as depicted by the blue color in Fig. 7. Suburban and exurban counties are projected to have large housing growth rates over the next 20 years, which is reasonable as urbanization in metro areas continues. There are noticeable differences in projected housing rates within the state of Texas. In the eastern half of the state, there is large projected growth around the state's major cities which aligns with recent trends.
Six of the 15 fastest-growing large cities in the US between 2010 and 2019 are located in Texas (U.S. Census Bureau, 2020b). However, housing inventory is projected to generally remain stagnant or decline in western Texas, which aligns with past trends of generally stagnant population and available jobs in the region (Texas Comptroller, 2020).



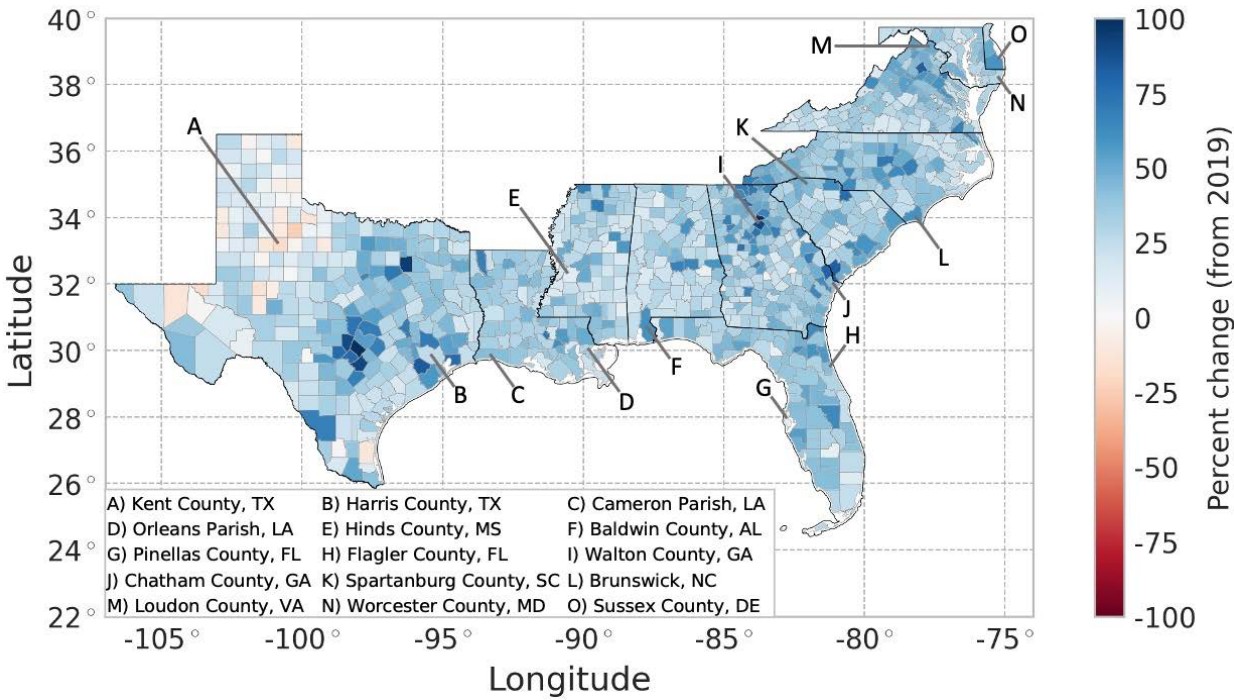

**Figure 7 Projected 20-year (2019–2039) percent change in housing units using the REACH20 model**

A comparison of the housing rates of change in the past 20 years (1999–2019) versus the next 20 years (2019–2039) allows for an analysis of housing growth acceleration or deceleration. The vast majority of counties (89.5 %) in the study area are expected to experience greater housing growth rates in the next 20 years (2019–2039), than in the past 20 years (1999–2039), as depicted by the green color in Fig. 8. These higher growth rates indicate that most counties need to carefully manage the rapid new home construction. Additionally, three out of four (75.4 %) counties in the region are expected to experience at least a 10 % change in housing rates in the projected 20 years versus the past 20 years. Two of five counties (38.2 %) in the region are expected to experience at least a 20 % change in housing rates between the two periods. This means that a simple linear extrapolation from the past 20 years will likely not provide an accurate projection of housing units.


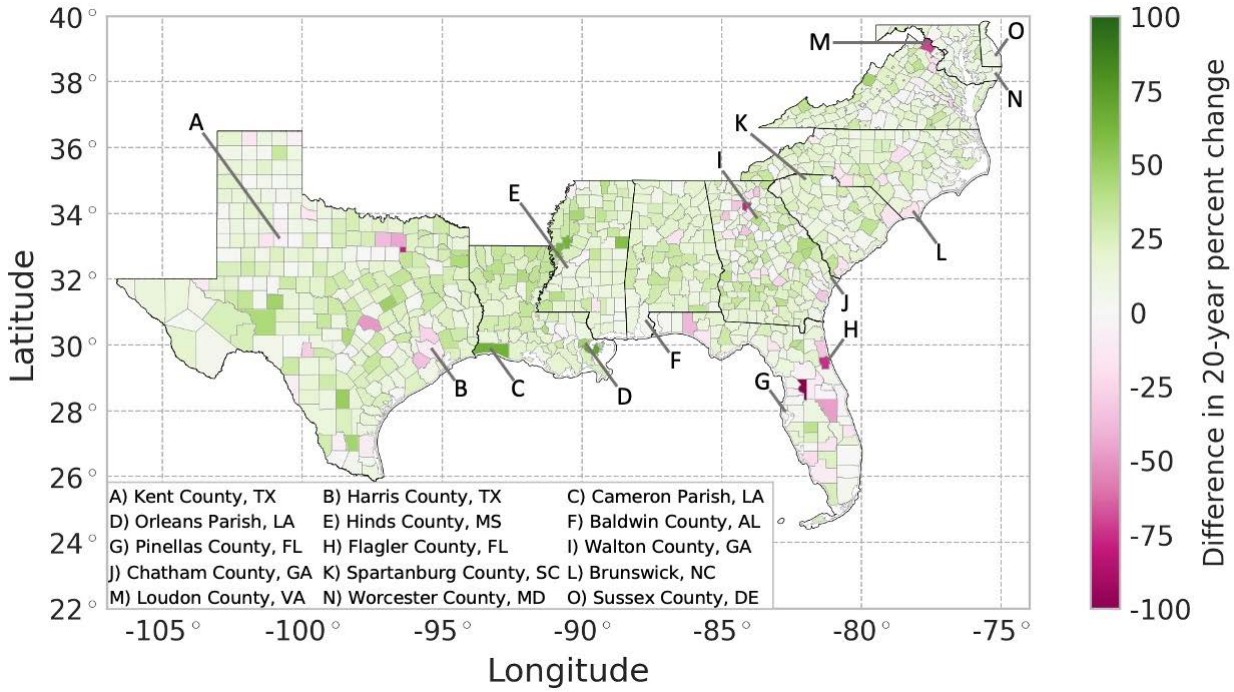

**Figure 8 Projected 20-year housing acceleration (Projected percent change in housing units between 2019 and 2039, minus the percent change of housing units between 1999 and 2019) using the REACH20 model**

A change in housing units over time also implies a change in housing density over time, often resulting in increased urbanization within a county. Fig. 9 shows the difference between the projected housing unit density in 2039 and the housing unit density in 2019. Most counties (71.4 %) are only expected to see a change of 10 housing units per $km^2$ or less in the next 20 years. However, one fifth of the counties (21.2 %) in the region are expected to experience an increase of 10 to 50 housing units per $km^2$, many of which are located along the Atlantic Coast. Notably, the vast majority of the counties along Florida's

coastline (74.5 %) are expected to experience an increase of 10 to 100 housing units per $km^2$. Of the coastal counties, Harris County is expected to experience the greatest increase in housing density, from 400 housing units per $km^2$ in 2019 to 510 housing units per $km^2$ in 2039. Areas of high density allow the possibility of more homes being affected by a single hurricane or other hazard event.



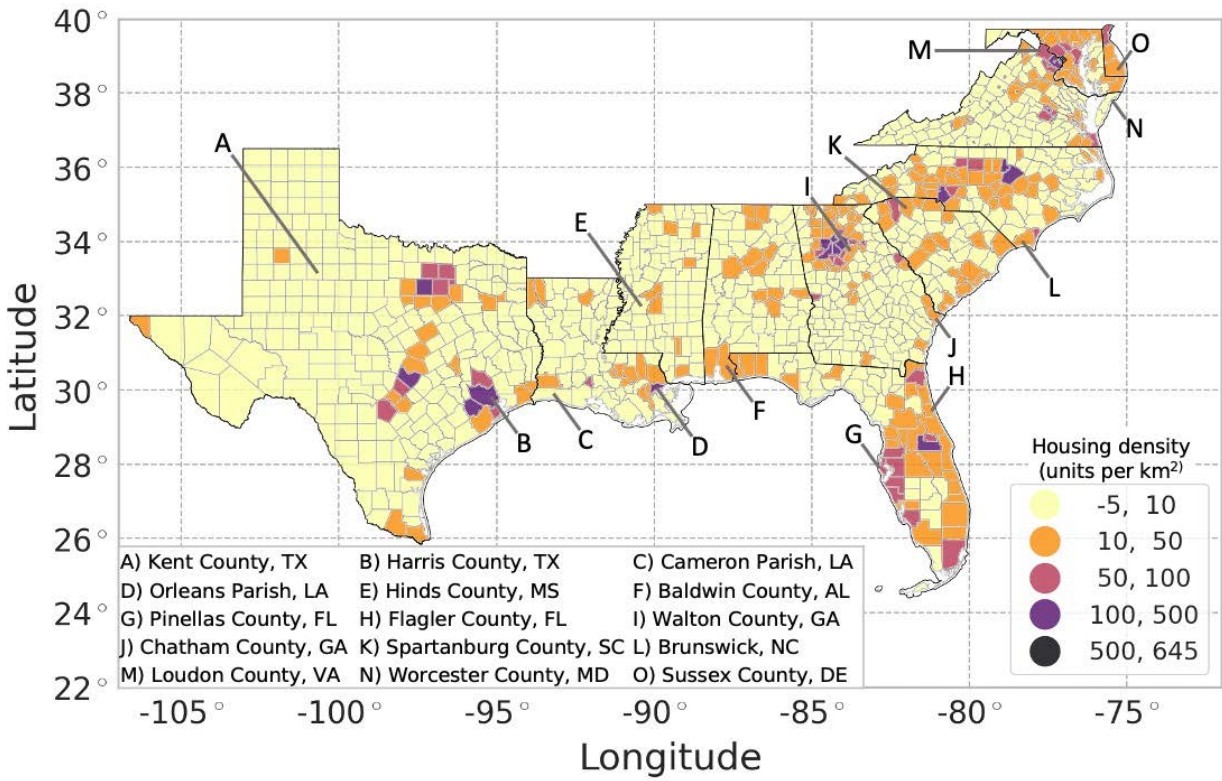


**Figure 9 Projected 20-year change in housing unit density (2039 housing unit density minus 2019 housing unit density) using the REACH20 model (units per km²)**

To investigate the projected number of housing units in more detail, a sample of 15 counties is identified (Fig. 9, and Fig. 10).

The 15 counties selected, which include one or two from each state in the study area (excluding Washington D.C.) and ten on the coast in hurricane-prone areas, were selected to illustrate some of the variability across counties. In five of the sampled counties (Kent County, TX, Harris County, TX, Flagler County, TX, Brunswick County, NC, and Loudon County, VA), the future housing trend (growing or shrinking) is expected to decelerate over the next 20 years, compared to the last 20 years. The two Louisiana parishes sampled (Fig. 10c), Cameron Parish and Orleans Parish (Fig. 10d), however, are examples of

exceptions that experienced significant shocks in the housing inventory due to hurricane impacts.





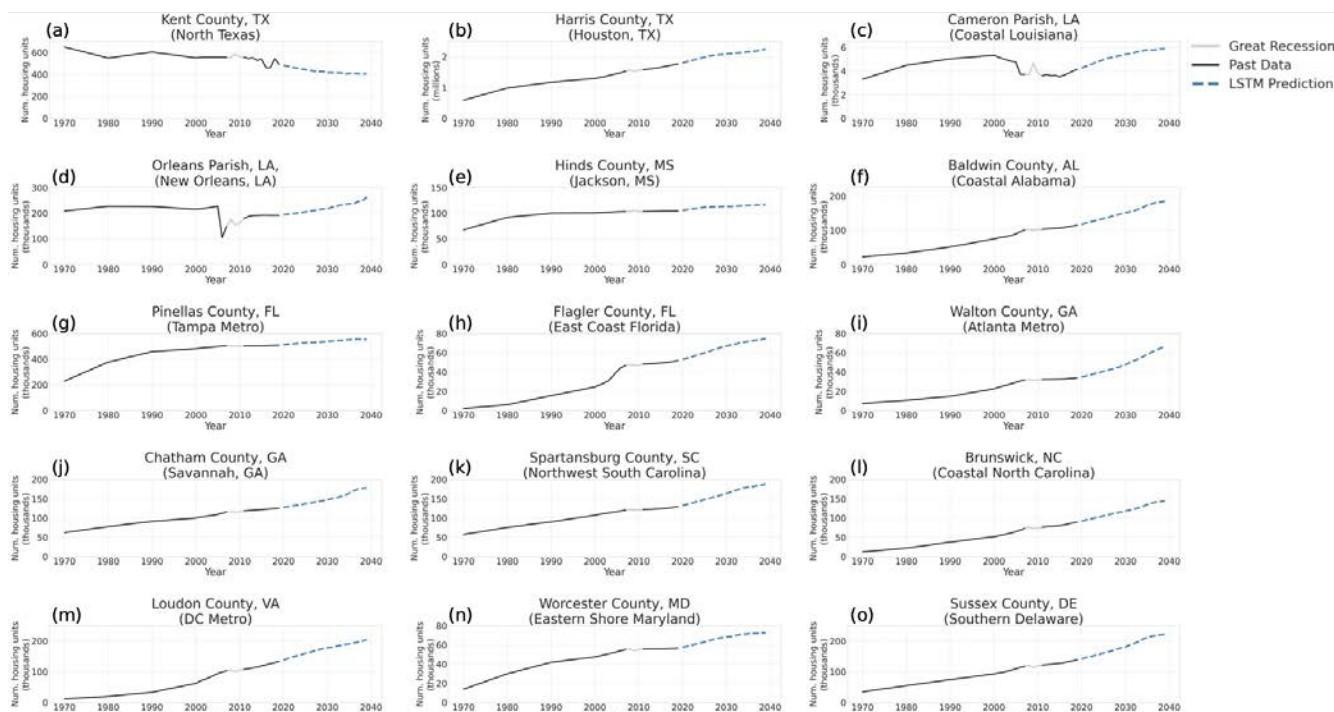

**Figure 10 Past and projected number of housing units for 15 Counties using the REACH20 model (note different scales)**

## 6.4 Implications for hurricane impacts and losses

The dynamics of the housing inventory also cause changes in a region's level of risk for multiple hazards, including hurricanes. Hurricane Harvey was a devastating Category 4 hurricane that made landfall on the Texas coast on August 25, 2017, affecting many counties in southeast Texas. Coastal counties experienced 130-mph winds, heavy rains, and large storm surges, while inland counties, particularly in the Houston, TX area, experienced massive amounts of rain over multiple days. Across the 62 counties affected by Hurricane Harvey, there was US$2.4 billion in residential property losses and US$7.5 billion in flood

insurance losses (Texas Department of Insurance, 2019). Using the proposed REACH20 model, if a hurricane of similar magnitude to Harvey hit that same Texas region 20 years from now, assuming the same 2017 distributed hazard and vulnerability profiles of newly built homes, the residential property losses for the entire region would be $3.2 billion, which is US$792 million (or 33 %) greater than the damage caused by Hurricane Harvey in 2017, assuming constant dollars. The total flood losses for the region would be even larger, totaling to $10.4 billion, which is approximately US$3 billion (or 40 %)

greater than Hurricane Harvey losses.

A closer examination of the projected housing growth rates across the subregions affected by Hurricane Harvey reveals that each subregion would experience a different magnitude of losses. The subregions analyzed align with the four areas identified


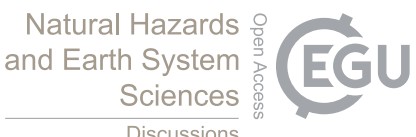

by the Texas Department of Insurance 2019 Report which documents insurance claims and losses from Hurricane Harvey in
the state of Texas. Using the recommended REACH20 model, it is expected that the projected 20-year housing growth rates
for each county in these areas will vary over the region (Fig. 11a) and the number of housing units will increase in each
subregion (Fig. 11b). For each subregion, the experienced residential property and flood losses from Hurricane Harvey and
the expected property and flood losses from a Hurricane Harvey-like event in 20 years are provided in Fig. 12. These values
are calculated by multiplying the loss values from Hurricane Harvey in each subregion by the expected 20-year housing growth
rate for each subregion.

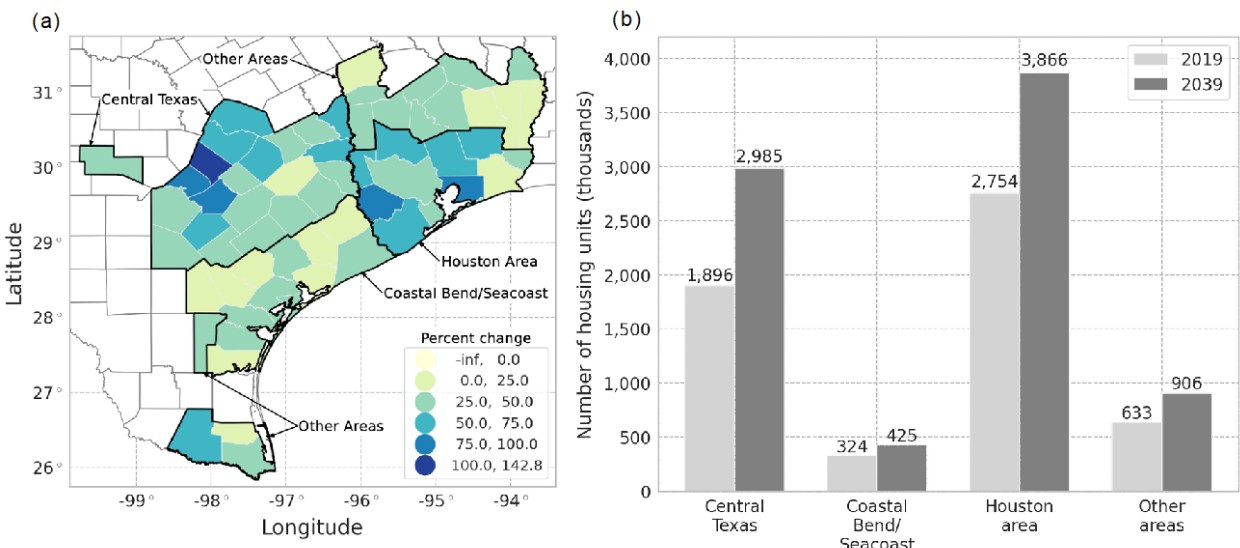

**Figure 11 (a) Projected 20-year (2019–2039) percent change in housing units in the Hurricane Harvey affected region using the REACH20 model (b) Projected housing units and housing growth rates for the Hurricane Harvey affected region using the**
**REACH20 model**





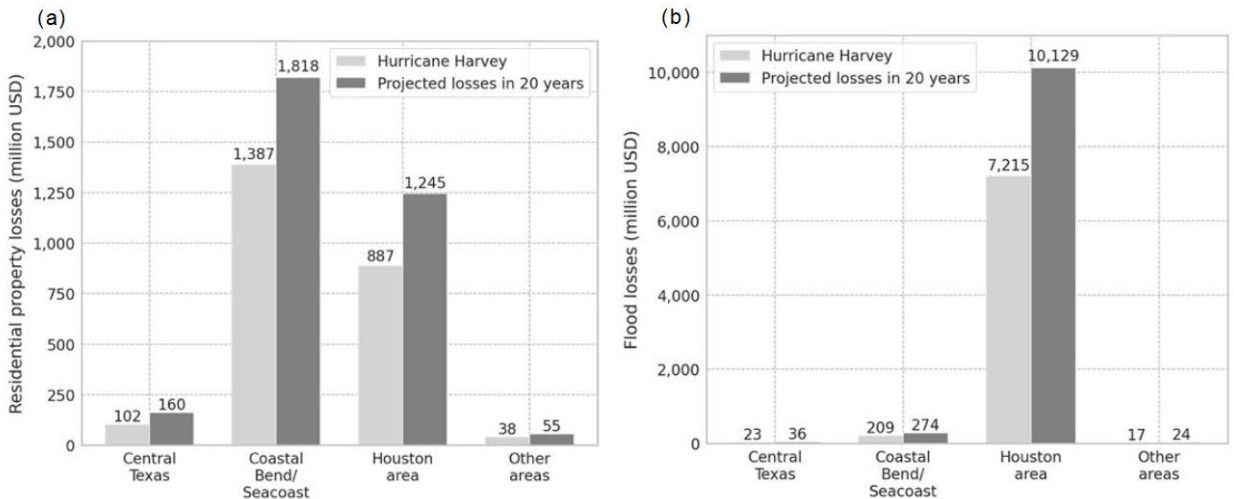

**Figure 12 (a) Estimated residential property loss due to hurricane impact across the Hurricane Harvey affected region over a 20-year period, assuming constant dollars (b) Estimated flood loss due to hurricane impact across the Hurricane Harvey affected region over a 20-year period, assuming constant dollars**


The area identified as the Coastal Bend and Seacoast Counties experienced the brunt of the wind force from Hurricane Harvey and accounted for the largest residential property losses (US$1.4 billion). Residential property losses account for the majority of damages due to high winds and include claims from homeowner's insurance, mobile homeowners' insurance, and residential dwelling insurance. The Coastal Bend area is expected to have the lowest housing growth rate of the region (31.0 %, or

approximately 100,000 more housing units), yet a similar-sized storm event hitting the same area in 20 years would result in an estimated US$431 million more losses than experienced in 2017, assuming constant dollars.

The area identified as the Houston Area and Southeast Texas experienced a massive amount of rainfall from Hurricane Harvey and accounted for the largest flood losses compared to other subregions (US$7.2 billion). The flood insurance losses reported

are caused by rising water or flood damages in residential or commercial structures and includes both federal and private flood insurance properties. The majority of flood insurance claims were for residential structures under the National Flood Insurance Program (NFIP). The Houston area is expected to have a sizeable housing growth rate of 40.0 %, equating to 1.1 million more housing units, over the next 20 years, which would cause a significant increase in expected flood losses for a similar-sized hurricane (US$2.9 billion).



## 7 Future work and conclusions

### 7.1 Limitations and future work

The recommended REACH20 model provides a first-of-its kind dataset of annual projected housing inventories for a multi-state region over a 20-year period that can be used to enhance hurricane risk models. Given the nature of the available data and complexity of the modeling method, there are limitations to note. For periods when data were only available at a decadal scale for certain variables, linear interpolations were made to produce an annualized dataset which could have introduced errors to the projection of housing units. Additionally, the data during the Great Recession (2008–2010) were removed because the model can neither predict nor account for large, unexpected exogenous shocks to the residential housing market. Additionally, the projected changes in housing units ultimately assume that past housing development behavior will carry into the future. However, housing demands have changed since the start of the COVID-19 pandemic and it is unclear how these changes are likely to affect future housing development trends. Climate change may also drive new behaviors in housing development patterns as risks due to sea level rise, intense storm events, wildfires, and excessive heat continues to increase. This study also had counties included in both the training and testing set because the model is only intended to be used for the designated study area. If the model were to be applied outside the study area, a review of holdout validation errors would be required. Lastly, neural network methods require a certain level of expertise and a significant effort to gather and standardize large quantities of data. Therefore, for applications only requiring quick estimates for changes in housing units, a simpler linear trend or ARIMA model may be adequate.

There is an opportunity to extend the housing unit projection work and estimate the likely distribution of housing unit types (e.g., single-family, multi-family, manufactured homes) in each county in the future. Researchers can also extend this work by estimating the likely location of the projected housing units within a given county, which would allow for a more granular estimate of hurricane impacts in a region. Additionally, researchers can evaluate potential policy mechanisms that can minimize the hurricane risk for a region while also incorporating the everchanging housing growth over time. Lastly, the provided housing unit projections can be applied to a variety of applications, including hurricane evacuation planning, hurricane risk mitigation, or general regional planning activities.

### 7.2 Conclusions

The recommended REACH20 model advances the field of hurricane risk modeling by producing the first known dataset of county-level annual housing inventory projections over a multi-decade period and multi-state region. It allows a dynamic building inventory to be included in hurricane risk models rather than using the conventional assumption of a static building inventory, thereby producing more realistic regional loss estimates. Additionally, the REACH20 model uses publicly-available
housing and demographic data and can therefore be easily applied to other regions of interest (see Sect. S2.3 of Supplemental Section for source code).

LSTM models outperformed linear trend and ARIMA models on all metrics; and the multivariate LSTM models outperformed
the univariate LSTM models, although when inclusion of additional feature variables meant fewer years of available data they did not lead to improved model performance. Applying spatial weighting by averaging a county's feature values with adjacent counties did not improve model results either. The REACH20 model includes 11 years of input data, 20 years of output data, 13 feature variables and a single target variable for 1,000 counties in the southeastern United States over 46 years of available data, resulting in 16,000 samples available to train and test the LSTM model. Using an 80/20 training/testing split, the 64,000
predicted data points in the testing set (3,200 samples in the testing set (16,000*0.2) and 20 predicted years), almost all (97.3 %) had absolute percent relative errors of less than one percent ($|H_{itk}|<1.0$), meaning the estimated number of housing units was no more than 1 % different than the actual number, errors that are acceptable for most practical purposes. The $\bar{H}$ remained less than 0.5 % for all 20 predicted time steps and errors were distributed evenly across the study region.

The REACH20 model suggests there will be significant increases in the housing inventories of the southeastern United States, thus increased expected hurricane losses. Of the 1,000 counties in the study area, 974 are expected to experience a growth in their housing inventory and 895 counties are expected to have greater housing growth in the next 20 years compared to the past 20 years. Translating to potential hurricane losses, if a Hurricane Harvey-type event hit southeastern Texas in 20 years, losses could increase by approximately 40 %, compared to the losses caused by Hurricane Harvey in 2017. Recognizing the
great expected hurricane losses, planners should prioritize mitigation and adaptation measures in the areas with high expected housing growth, thereby decreasing future societal impacts and financial losses.

**Supplemental Section**

Information about the data sources, data processing, modeling methods, and analysis methods is provided in the Supplemental Section available on the DesignSafe Data Depot under Project PRJ-3303 (Williams and Davidson, PENDING).

**Code availability**

All code is available on the DesignSafe Data Depot under Project PRJ-3303 and supporting documentation is available in the Supplemental Section (Williams and Davidson, PENDING).





## Data availability

All data consumed and produced is available on the DesignSafe Data Depot under Project PRJ-3303 and supporting documentation is available in the Supplemental Section (Williams and Davidson, PENDING).

## Author contribution

CW was involved in the data curation, formal analysis, software implementation, and preparation of the visualizations; RD supervised the project at large; CW and RD were involved in the investigation, methodology, validation, and the original preparation and writing of the draft; RD, LN, JT, MM, and JK were involved in conceptualization, funding acquisition, project

administration, and supplying resources for the project; CW, RD, LN, JT, MM, JK assisted in the final review and editing of the text.

## Competing interests

The authors declare that they have no conflict of interest

## Acknowledgements

This material is based on work supported by the National Science Foundation under award #1830511. The statements, findings, and conclusions are those of the authors and do not necessarily reflect the views of the National Science Foundation.

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
