# Peer review of "Regional county-level housing inventory predictions and the effects on hurricane risk"

_Natural Hazards and Earth System Sciences, 2021_

## Referee Comment (RC1)

**Review of:** Regional county-level housing inventory predictions and the effects on hurricane risk (nhess-2021-335)

**General Comments:**

Overall, I enjoyed the manuscript and think the work is very solid. I am especially appreciative of the well-written manuscript and great figures/tables. The technical discussion of their model and results is also very good. The manuscript does suffer a bit when it comes to the background and literature, though. The authors do this great work but gloss over many prior studies that have asked similar questions. Below I make several recommendations to improve the reach of this manuscript and help place it in the context of existing studies that have looked at similar issues.

**Recommendation:** Revisions Requested

**Specific Comments:**

- This research describes the plethora of studies centered on "Expanding Bull's-Eye Effect" (See: https://chubasco.niu.edu/ebe.htm) As such, there needs to be discussion included on how this manuscript furthers the knowledge related to hazard exposure changes over time. I encourage the authors to examine these manuscripts, especially Freeman and Ashley (2017), for additional studies (e.g., works by Preston et al.) to support their findings. Recommend including in Section 2.3

- Line 115: This is not true. Most of these studies that have used SERGoM and ICLUS have their housing unit projections controlled by historical (or climate change storyline projections) county-level enumerations of housing unit growth rates. This reasoning is weak and built on a shaky foundation, at best. For instance, Freeman and Ashley (2017) examined multiple states and metro areas (made up of multiple counties).

- There needs to be more discussion on how the housing growth models used herein differ compared to other methods (e.g., dasymetric). Is the method presented herein "better", or is it just "different"? Either way, more discussion is needed beyond ML and CNN models (Section 2.1)

- Section 4.1: Why was an OLS regression used over other counterparts. There needs to be at least some discussion on it's benefits and reason for selection.

- The results section could be cut down a bit for brevity. I like the technical discussions but think it could be condensed quite a bit or moved into supplemental material. My reasoning for bringing this up is so that they can add a section in the discussion portion of the manuscript that allows them to compare their results against others. This provides context and potential areas of future improvement. It also may highlight where the author's methods are superior.

**Technical Corrections:**

- Recommend removing paragraphs and sentences that start with "Table X shows…" Figure Y illustrates…" etc. For example, line 121-124 is is caption material, not text material. Parenthetical referencing will remove the "fluff" from the text and help the manuscript flow much better.

- Line 145: Already defined acronyms/initializations prior.

---

## Referee Comment (RC2)

**General comments**

The authors implemented a neural network based model and simulated annual housing inventory changes. The modeling of inventory changes helps to better assess hurricane risks. The proposed methods have both academic and practical merits. The methodologies are well-documented and validated against benchmark models (a time-series and a linear regression model). The authors addressed a critical line of inquiries, particularly during the era of unpredictable climate change trends and vulnerable human society systems in response to external shocks. With these being said, a few comments below may help with the refinement of the manuscript and further progress to the next step.

Line 35: It is interesting to state the current research gaps here concisely before jumping into contributions, which can provide research the context. The gaps and difficulties in employing an updated housing inventory into current risk assessment frameworks can be discussed in more details in the literature review section.

Line 40: This second contribution appears to be a little vague to me. In its current form, this statement sounds more like a methodological summary rather than a contribution highlight.

Line 55: Please shorten the reviews of land use and change, which is less relevant to the article. Yet, you may want to expand Section 2.2, Housing economics. Various drivers of housing development should interest readers who want to learn more about forces underlying housing inventory changes.

Line 129: Why is county chosen as a unit of analysis (UOA)? How can your selection of UOA affect modeling outcomes? And is this unit applicable to other areas with different geographical and/or administrative context.

Line 142: Change the title to "model specifications"? Model types appears to be confusing.

Line 151. Maybe add an equation to define the mentioned model specifications

Line 170 through 175. Please cite references for these statements. And please use a chart or conceptual mathematical expression to illustrate how LSMT and neutral networks (in the context of housing change modeling) work. This may non-computer science experts better understand the concept.

Line 188: Please combine section 4 and 5 into a methods section. Currently, these two sections appear to be disconnected.

Line 395: Figure 5 is too small.

Figure 10: Re-arrange sub-figures to make it easier to read x and y labels.

---

## Author Comment (AC1)

**Response to comments:** Regional county-level housing inventory predictions and the effects on hurricane risk (nhess-2021-335)

RC1

**General Comments:**

Overall, I enjoyed the manuscript and think the work is very solid. I am especially appreciative of the well-written manuscript and great figures/tables. The technical discussion of their model and results is also very good. The manuscript does suffer a bit when it comes to the background and literature, though. The authors do this great work but gloss over many prior studies that have asked similar questions. Below I make several recommendations to improve the reach of this manuscript and help place it in the context of existing studies that have looked at similar issues.

> Thank you for your supportive comments. We have carefully reviewed your comments regarding the background and literature section and our response is below.

**Specific Comments:**

This research describes the plethora of studies centered on "Expanding Bull's-Eye Effect" (See: https://chubasco.niu.edu/ebe.htm) As such, there needs to be discussion included on how this manuscript furthers the knowledge related to hazard exposure changes over time. I encourage the authors to examine these manuscripts, especially Freeman and Ashley (2017), for additional studies (e.g., works by Preston et al.) to support their findings. Recommend including in Section 2.3.

> Thank you for the recommended references. We propose modifying Section 2.3 as shown, so it introduces the expanding bull's-eye effect earlier in the paragraph alongside the relevant works of Ashley and Strader relating to changing building exposure and natural hazard risk.

> *"There is a limited group of studies that evaluate a society's changing exposure to natural hazard risk over time. Davidson and Rivera (2003) use population projections and headship rate data to predict the number, location, and types of housing units per census tract in a region at 5-year intervals between 2000 and 2020. The results were later used in a hurricane risk study for North Carolina (Jain and Davidson, 2007). Multiple studies have evaluated the "expanding bull's-eye effect, a phenomenon in which the expansion of a metropolitan area's urban, suburban, and exurban regions leads to an increase in the area's natural hazard risk, due to the expanding footprint of the built environment (Ashley et al., 2014). Ashley and Strader (2016) explored the expanding bull's-eye effect on tornado impacts in the contiguous US as a whole, as well as five multi-state regions within the US between 1950 and 2010 at decadal intervals by utilizing the housing density data produced by the CA-based Spatially Explicit Regional Growth Model (SERGoM) (Theobald, 2005). Strader et al. (2015) used SERGoM and the US EPA's Integrated Climate and Land Use Scenarios (ICLUS) to forecast exposure to*

*volcanic hazard in the Northwest US at a decadal scale between 2010 and 2100 under five scenarios. Similarly, Freeman and Ashley (2017) used SERGoM to forecast hurricane risk in the US for the same time interval under two hurricane scenarios, and Strader et al. (2018) explored how ten different land development patterns would impact a region's tornado risk. Chang et al. (2019) studied the effect of urban development patterns on future flood risk or earthquake risk in the Vancouver regions for the year 2041 under three prescribed development scenarios—status quo, compact, and sprawl. Song et al. (2018) compared three ML methods to predict the land use change in Bay County, Florida in 2030 and evaluated the risk due to sea level rise under two growth rates and two policy scenarios. Hauer et al. (2016) used a modified version of the Hammer method (Hammer et al., 2004) to predict the number of people at risk of sea level rise per census block, based on decadal housing estimates for the coastal areas of the contiguous US, between 2010 and 2100 under five development scenarios. Sleeter et al. (2017) used a CA model to evaluate changes in land cover and the effect on tsunami risk in the US Pacific Northwest at annual increments between 2011 and 2061. Keenan and Hauer (2020) compared 30-year population projections in Puerto Rico with planned hurricane recovery and resiliency investments, finding an overestimation of future fiscal and infrastructure needs compared to the projected decline in population."*

Line 115: This is not true. Most of these studies that have used SERGoM and ICLUS have their housing unit projections controlled by historical (or climate change storyline projections) county-level enumerations of housing unit growth rates. This reasoning is weak and built on a shaky foundation, at best. For instance, Freeman and Ashley (2017) examined multiple states and metro areas (made up of multiple counties).

We agree that that sentence was not worded well. We propose removing it and replacing it with the following paragraph in Sect 2.3.

*"This paper contributes to this literature by similarly modeling the effect of changing exposure on natural disaster risk over time. In general, the best method will depend on the specific intended use and required output, which together with data availability, determine the most appropriate target metric and spatial and temporal units of analysis and scope. With a focus on hurricane risk, in this paper we aim to develop annual forecasts of the number of housing units in each county in the hurricane-prone US for the next two to three decades. The aforementioned studies that similarly include county-level housing unit forecasts (although with varied overall aims) compute those forecasts by obtaining population projections from private organizations or public agencies and by applying a constant housing unit per population ratio to produce county-level housing projections in five- or ten-year increments (Hauer et al., 2016; Ashley and Strader, 2016; Strader et al., 2015; Freeman and Ashley, 2017; Strader et al., 2018; Sleeter et al., 2017; Davidson and Rivera, 2003). In this study, we examine whether accurate annual county-level housing unit forecasts are possible using machine learning with a housing unit target variable and land and socio-economic features."*

There needs to be more discussion on how the housing growth models used herein differ compared to other methods (e.g., dasymetric). Is the method presented herein "better", or is it just "different"? Either way, more discussion is needed beyond ML and CNN models (Section 2.1)

The best model depends to some extent on the specific intended use and data availability. For example, in some cases, if population projections are available and the resulting errors are acceptable, the product of constant housing unit per population ratios and population projections may be most appropriate. In others, if population projections are not available or more precision in the forecast is required, a different method may be preferred. We have compared three different common model types (linear, ARIMA, LSTM) to give an idea of the tradeoff between simplicity and accuracy. Of course other model types exist. Unfortunately, we were not able to obtain the required data to conduct a fair comparison with the approach based on population projections and constant population to housing unit ratios. As an effort in that direction, however, we propose adding the text below to the second paragraph in Section 6.2 in which we evaluate the recommended LSTM model. We do not believe that dasymetric modeling in particular is applicable for our aims because we were not producing sub-county housing unit estimates.

*"Furthermore, the population projection method provided by Hauer (2019) for all US counties produce aggregated relative errors of 0.9% to 3.6% over a 15-year projection period, while the recommended model in this study produces average absolute relative errors of less than 0.5% over a 20-year projection period. This suggests that if a static housing unit per population ratio was applied to the population estimates produced by Hauer (2019), as is done in other studies evaluating natural hazard risk in the context of a changing housing inventory (Hauer et al., 2016; Ashley and Strader, 2016; Strader et al., 2015; Freeman and Ashley, 2017; Strader et al., 2018; Sleeter et al., 2017; Davidson and Rivera, 2003), these housing estimates would likely be less accurate than those produced by the recommended REACH20 model."*

Section 4.1: Why was an OLS regression used over other counterparts. There needs to be at least some discussion on it's benefits and reason for selection.

We agree that it is unclear why OLS regression was chosen in the model comparison and propose splitting the sentence beginning on Line 147 (in the original document) within Section 4 into the following four sentences:

*"Linear trend models were included in the model comparison as a baseline because they are commonly used in forecasting applications, are quick to implement, and are easy to interpret. ARIMA models were tested because they are easy to use, commonly applied across a range of disciplines, and interpretable. LSTM models were considered for their ability to handle large quantities of spatial and temporal data and produce small errors.*

*These three models were ultimately chosen to compare the tradeoffs between model simplicity and model accuracy; if the linear or ARIMA models produce errors in the same range as the LSTM models, then these simpler models may be recommended for housing projections."*

The results section could be cut down a bit for brevity. I like the technical discussions but think it could be condensed quite a bit or moved into supplemental material. My reasoning for bringing this up is so that they can add a section in the discussion portion of the manuscript that allows them to compare their results against others. This provides context and potential areas of future improvement. It also may highlight where the author's methods are superior.

A primary purpose of this study is to understand whether recurrent neural network models, like LSTMs, can be used for housing projection applications. No known study has used ML methods for annual county-level housing projections and the paper provides insights on the intricacies required when applying an LSTM model (e.g., which time lengths should be used for inputs and outputs? Which features should be used? Does the inclusion of spatial weighting impact results?). This information is meant to support future researchers who wish to use recurrent neural networks for their changing built environment modeling, which is unavailable in the existing literature.

**Technical Corrections:**

Recommend removing paragraphs and sentences that start with "Table X shows…" Figure Y illustrates…" etc. For example, line 121-124 is is caption material, not text material. Parenthetical referencing will remove the "fluff" from the text and help the manuscript flow much better.

Lines 121-124, 133, 191, 256, 370, 384, 399, 406, 420, 431, 473 have been adjusted to remove "Table/Figure X shows…" Relevant information has been added to the associated figure/table captions.

Line 145: Already defined acronyms/initializations prior.

The acronyms have been removed from Line 145.

---

## Author Comment (AC2)

**Response to comments:** Regional county-level housing inventory predictions and the effects on hurricane risk (nhess-2021-335)

RC2

**General comments:**

The authors implemented a neural network based model and simulated annual housing inventory changes. The modeling of inventory changes helps to better assess hurricane risks. The proposed methods have both academic and practical merits. The methodologies are well-documented and validated against benchmark models (a time-series and a linear regression model). The authors addressed a critical line of inquiries, particularly during the era of unpredictable climate change trends and vulnerable human society systems in response to external shocks. With these being said, a few comments below may help with the refinement of the manuscript and further progress to the next step.

> Thank you for your supportive comments. Our responses are provided below.

Line 35: It is interesting to state the current research gaps here concisely before jumping into contributions, which can provide research the context. The gaps and difficulties in employing an updated housing inventory into current risk assessment frameworks can be discussed in more details in the literature review section.

> We agree that it may be too early to state the contributions prior to discussing the research gaps, however we would prefer to contain the discussion about research gaps within the literature review section, rather than introducing research gaps in the introduction section. Therefore, we have changed line 35 to say:
>
> *"…paper has two outcomes. First, using…"*
>
> which allows us to set up the direction of the paper without getting in the weeds of the existing literature.

Line 40: This second contribution appears to be a little vague to me. In its current form, this statement sounds more like a methodological summary rather than a contribution highlight.

> We believe the usage of "outcomes" instead of "contributions" in line 35 resolves this issue.

Line 55: Please shorten the reviews of land use and change, which is less relevant to the article. Yet, you may want to expand Section 2.2, Housing economics. Various drivers of housing development should interest readers who want to learn more about forces underlying housing inventory changes.

> We prefer to retain the land use and population projection literature section because most research that explores the interface of natural hazards and a changing built environment are rooted in the land use and population change modeling methods. In particular, we would like to retain the section that distinguishes machine-learning (ML) methods and cellular automata (CA) methods used in land change modeling because many of the natural hazard application studies use ML or CA methods. Additionally, the predictor variables listed in Table 1 represent drivers of housing development that we believe will interest readers who want to learn more about the forces underlying housing inventory changes.

Line 129: Why is county chosen as a unit of analysis (UOA)? How can your selection of UOA affect modeling outcomes? And is this unit applicable to other areas with different geographical and/or administrative context.

> A sentence has been added at the beginning of Sect. 3 to clarify the choice of counties as the UOA. The revised section of Sect. 3 is provided below, where the additional sentence is highlighted.
>
> *"Modeling the annual changes in the number of housing units for 1,000 counties over a 10-, 20-, or 30-year time horizon requires a dataset of annual county-level data for more than 10 years for all counties in the study area. Counties were chosen as the unit of analysis, opposed to census tracts, block groups, or a grid cells, because county boundaries rarely change over a multi-decade period and data is available at the county-level over multiple decades for most of the predictors in Table 1. Of the 32 predictors identified as potential predictors of new housing construction, 25 (indicated by "2" in Table 1) had county-level data available for more than 10 years and were considered for this study."*
>
> Note also that the target variable is annual percent change in the number of housing units (as opposed to number of housing units), which reduces the effect of county size.
>
> Any UOA can be utilized in different geographical and/or administrative contexts as long as there is annual data available for the 13 features in the REACH20 model between 1971 and 2019.

Line 142: Change the title to "model specifications"? Model types appears to be confusing.

> We prefer to retain "Model types" as the header for Section 4. Linear, ARIMA, and LSTM models are distinctly different forecasting methods, i.e. model types. Modeling specifications sounds like it refers to specific instances of each model type.

Line 151. Maybe add an equation to define the mentioned model specifications

> "y = mx + b" has been added to Line 153 to clarify the simple linear trend model type.

Line 170 through 175. Please cite references for these statements. And please use a chart or conceptual mathematical expression to illustrate how LSMT and neutral networks (in the context of housing change modeling) work. This may non-computer science experts better understand the concept.

> Given that the paper is already very long, we purposefully provided three papers in line 178 for readers to reference if they are interested in learning more about the mechanics of LSTM models, rather than explaining the intricate inner workings of LSTM models.

Line 188: Please combine section 4 and 5 into a methods section. Currently, these two sections appear to be disconnected.

> In our field, it is common to first present the modeling methods used in a general format (Section 4) before introducing the application of the modeling structure for our problem at-hand (Section 5). Therefore, we would prefer to retain a separation between these two sections. However, if the editor would like us to combine these two sections, we will do so.

Line 395: Figure 5 is too small.

> Figure 5 has been adjusted to match the size of similar maps in Figures 7, 8, and 9.

Figure 10: Re-arrange sub-figures to make it easier to read x and y labels

> Large labels have been added to the x- and y-axis of Figure 10 to make the information easier to read.